# Glucose Biosensor Based on Glucose Oxidase Immobilized on BSA Cross-Linked Nanocomposite Modified Glassy Carbon Electrode

**DOI:** 10.3390/s23063209

**Published:** 2023-03-17

**Authors:** Yang-Yang Li, Xin-Xin Ma, Xin-Yan Song, Lin-Lin Ma, Yu-Ying Li, Xin Meng, Yu-Jie Chen, Ke-Xin Xu, Ali Akbar Moosavi-Movahedi, Bao-Lin Xiao, Jun Hong

**Affiliations:** 1School of Life Sciences, Henan University, Kaifeng 475000, China; 2Institute of Biochemistry and Biophysics, University of Tehran, Tehran 1417614418, Iran

**Keywords:** glucose oxidase, biosensor, hydroxyl fullerenes, multi-walled carbon nanotubes, bovine serum albumin

## Abstract

Glucose sensors based blood glucose detection are of great significance for the diagnosis and treatment of diabetes because diabetes has aroused wide concern in the world. In this study, bovine serum albumin (BSA) was used to cross-link glucose oxidase (GOD) on a glassy carbon electrode (GCE) modified by a composite of hydroxy fullerene (HFs) and multi-walled carbon nanotubes (MWCNTs) and protected with a glutaraldehyde (GLA)/Nafion (NF) composite membrane to prepare a novel glucose biosensor. The modified materials were analyzed by UV-visible spectroscopy (UV-vis), transmission electron microscopy (TEM), and cyclic voltammetry (CV). The prepared MWCNTs-HFs composite has excellent conductivity, the addition of BSA regulates MWCNTs-HFs hydrophobicity and biocompatibility, and better immobilizes GOD on MWCNTs-HFs. MWCNTs-BSA-HFs plays a synergistic role in the electrochemical response to glucose. The biosensor shows high sensitivity (167 μA·mM^−1^·cm^−2^), wide calibration range (0.01–3.5 mM), and low detection limit (17 μM). The apparent Michaelis–Menten constant $K_{m}^{\mathrm{app}}$ is 119 μM. Additionally, the proposed biosensor has good selectivity and excellent storage stability (120 days). The practicability of the biosensor was evaluated in real plasma samples, and the recovery rate was satisfactory.

## 1. Introduction

Diabetes is a chronic disease caused by insulin deficiency and hyperglycemia. Attention to the blood glucose level is an indicator for the diagnosis and treatment of diabetes because of its multiple complications affects the normal life of many of diabetics [1,2,3]. The traditional techniques for detecting glucose (Glu), for example, colorimetry [4], fluorescence [5], and high-performance liquid chromatography [6] have disadvantages of sensitivity to interfering substances in samples, large sample size, and being time-consuming. Thus, the electrochemical method has attracted extensive attention because it complements the shortcomings of traditional methods and exhibits fast detection, simplicity, low cost, and excellent sensitivity [7,8,9,10]. Additionally, the biosensor based on glucose oxidase (GOD) develops rapidly due to its good selectivity and high sensitivity.

However, the signal transmission between the enzyme and the electrode is difficult due to the large structure of the enzyme molecule. In recent years, nanomaterials have been widely used in the design of electrochemical sensors due to their inherent nano effect, good electrical conductivity, and high catalytic activity. Examples include multi-walled carbon nanotubes (MWCNTs) [11,12,13], platinum [14] and gold nanoparticles [15,16], reduced graphene oxide [17,18,19,20], fullerenes [21], Fe_3_O_4_ nanoparticles [22,23], etc. Among them, fullerene is a carbon-based nano material with a spherical three-dimensional structure. Its mechanical stability, high electron transfer ability, and inert behavior have aroused the interest of researchers [24]. Hydroxyl fullerenes (HFs) were modified with functional hydroxyl groups to make them hydrophilic in nature, which helped them bind to proteins into complexes and to protect proteins, thus successfully being applied in biosensors [25,26]. Gao et al. reported that GOD was immobilized on HFs-modified glassy carbon electrode (GCE) to detect Glu [27], but the sensitivity was low. It was reported that MWCNTs have become one of the important nano materials of electrochemical sensors because of their high conductivity, large specific surface area, and outstanding sensitivity [28,29]. In this work, MWCNTs were introduced to enhance the biosensor sensitivity. However, the hydrophobicity of MWCNTs are very strong and have poor biocompatibility, which affects the performance of the biosensor. It was reported that bovine serum albumin (BSA) is a globular protein in plasma and has been used in the construction of biosensors due to its non-toxicity, biocompatibility, and non-immunogenicity [30]. For example, He et al. reported that GOD was immobilized on the composite nanoparticles based on gold nanoparticles/BSA/Fe_3_O_4_ to detect Glu and proposed that the shell with BSA supplied a biocompatible environment for GOD and helped to improve the activity of immobilized GOD [31]. The Glu biosensor was prepared on alumina mem-branes/Pt/polypyrrole nanotube arrays by cross-linking GOD with BSA and glutaraldehyde (GLA). Palod et al. proved that cross-linked fixation of GOD could improve the overall performance of biosensors such as sensitivity and shelf life [32]. Therefore, BSA was introduced to better fix GOD on MWCNTs-HFs to improve the biocompatibility of MWCNTs and the performance of the biosensor.

In this work, BSA was first used to cross-link and fix GOD onto HFs-MWCNTs nanocomposites, and then a layer of Nafion (NF)/GLA composite membrane was modified to protect the electrode. Finally, a Glu biosensor was constructed for the detection of rat plasma Glu concentration. Here, HFs was introduced to protect the conformation and properties of GOD [33], MWCNTs were used to enhance the electrical signal and sensitivity, and BSA was added to regulate the biocompatibility of the composite. MWCNTs, HFs, and BSA play a synergistic role in Glu molecular recognition. The modified materials were characterized by cyclic voltammetry (CV), UV-visible spectrophotometry (UV-Vis), and transmission electron microscopy (TEM). In addition, the test method and other factors, for instance, enzyme concentration, anti-interference ability, pH, stability, and so on, were systematically studied. Finally, the optimized biosensor successfully identified Glu molecules and determined Glu concentration in rat plasma. Thus, the biosensor has great potential in the measurement of human blood Glu concentration, clinical diagnosis, and management of diabetes.

## 2. Materials and Methods

### 2.1. Reagents and Materials

NF, GOD (EC 1.1.3.4, from *Aspergillus niger*), HRP (EC 1.11.1.7, type VI-A), guaiacol, sodium disodium hydrogen phosphate (Na_2_HPO_4_·12H_2_O), and dihydrogen phosphate (NaH_2_PO_4_·2H_2_O) were purchased from Sigma-Aldrich (Shanghai, China). HFs, MWCNTs, and BSA were obtained from Bucky (Houston, TX, USA), Shenzhen Nanotech Port Co., Ltd. (Shenzhen, China), and Shanghai Baoman Biotechnology Co., Ltd. (Shanghai, China), respectively. Vitamin B_1_ (Vit B_1_), vitamin C (Vit C), and sodium chloride (NaCl) were purchased from Beijing Dingguo Biotechnology Co., Ltd. (Beijing, China), Beijing Aoboxing Biotechnology Co., Ltd. (Beijing, China) and Tianjin Deen Chemical Reagent Co., Ltd. (Tianjin, China), respectively. GLA (0.1%) was obtained from Aladdin Reagents Co., Ltd. (Shanghai, China). All reagents in this study used without further purification. Millipore Milli-Q water (18 MΩ cm) was used in this work.

### 2.2. Apparatus and Measurements

Electrochemical experiments were conducted on CHI660E electrochemical workstation (CH Instruments, Austin, TX, USA). In a traditional three-electrode electrochemical cell, a saturated Ag/AgCl electrode was reference electrode, a modified GCE (diameter 3.0 mm) was working electrode, a platinum wire was counter electrode. Electrochemical and electrochemical catalytic determination were carried out respectively in N_2_-saturated and air bubbling (30 min, 300 mL·min^−1^) 50 mM, pH 6 phosphate-buffered solution.

TEM images of MWCNTs and HFs were collected by TEM (JEM2100, JEOL, Tokyo, Japan) at 200 KV.

The effects of MWCNTs, HFs, and BSA on the catalytic activity of GOD were studied using an ultraviolet and visible spectrophotometer (UV-vis spectrophotometer) (Evolution 220, Thermo, Shanghai, China) [34].

### 2.3. Preparation of Modified Electrode

Before fixing GOD, the surface of GCE was mechanically polished to the mirror surface with 1, 0.3, and 0.05 μm aluminum oxide aqueous suspension successively. Next, the GCE was treated ultrasonically with double ultrapure water and 75% ethanol for 10 min. Afterwards, the GCE was placed in a drying tower for drying [35].

Steps of modified GCE: firstly, 4 μL of GOD (10 mg·mL^−1^) and 2 μL of HFs (4 mg·mL^−1^) were mixed and then next mixed with 1 μL of BSA (1%), and then 4 μL of BSA-HFs-GOD mixture was mixed with 2 μL of MWCNTs (4 mg·mL^−1^). Subsequently, 5 μL of the composite solution was taken out and dripped onto the surface of the GCE and dried in a refrigerator at 4 °C. Lastly, the 2.5 μL NF-GLA complex (1:1 volume ratio) was immediately dropped on the GCE for protection after mixing. Figure 1 showed preparation process of modified electrode.

### 2.4. Sample Preparation

The requirements and experimental use of the animals were reviewed and approved The Biomedical Research Ethics Sub-Committee of Henan University (HUSOM2021-198). Glu plasma samples were prepared: firstly, the blood was obtained through orbital vein after mild anesthesia of rats [36]. Then, the blood of rats was combined with anti-coagulant and was centrifuged at 3000 r·min^−1^ for 30 min to obtain the plasma supernatant. Afterwards, plasma samples were frozen at −20 °C until used.

## 3. Results and Discussion

### 3.1. Characteristics of Modified Materials

The morphologies of MWCNTs and HFs were obtained by TEM. MWCNTs were curved tubes with a diameter of about 10–20 nm. According to the literature, the size of monomolecular HFs is about 1 nm [37]. Figure 2B showed the HFs were a near-spherical shape with an average size of about 20 nm, which may be due to the fact that HFs had cohesive hydrogen bonds and were easy to aggregate in aqueous solution [38,39,40]. HFs aggregates can form complexes with GOD [27], which may promote the electron transfer of protein sites and the conductivity of HFs and improve the catalytic ability of GOD.

UV-vis was used to detect the initial catalytic reaction rates (ICRR) of GOD in the presence of BSA (BSA/GOD), HFs (HFs/GOD), and MWCNTs (MWCNTs/GOD) to study the effects of BSA, HFs, and MWCNTs on the catalytic activity of GOD. The detection conditions were similar to our previous study [27]: HRP, GOD, guaiacol, and Glu were added to phosphate−buffered solution to start the reaction. Test wavelength and temperature were 470 nm and 25 °C, respectively. The initial oxidation rate of guaiacol was determined by the concentration of colored product (teraguaiacol, ε_470nm_ = 26.6 mM^−1^·cm^−1^). Afterwards, ICRR of the GOD could be converted according to the formation rate of teraguaiacol, and the activity of GOD could be obtained. The relative reaction formula can be expressed by Equations (1) and (2) [34]:(1)$${\mathrm{Glu}+O}_{2}\;\to {\;\mathrm{Gluconolactone}+H}_{2}O_{2}$$
(2)$${4\;H}_{2}O_{2}+4\;\mathrm{Guaiacol}\;\overset{\mathrm{HRP}}{\to }{\;\mathrm{Teraguaiacol}+8\;H}_{2}O$$

In Figure 3, the slopes of GOD, GOD-BSA, GOD-HFs, and GOD-MWCNTs had little change. The ICRR of GOD, GOD-BSA, GOD-HFs, and GOD-MWCNTs were all converted to 0.2 μM s^−1^. Thus, BSA, HFs, and MWCNTs hardly affect the ICRR of GOD, which may be due to the good biocompatibility of BSA, HFs, and MWCNTs with GOD.

### 3.2. Electrochemical Studies

Figure 4 shows the CVs behavior of different modified GCEs in N_2_-saturated 50 mM pH 6 phosphate-buffered solution with a scan rate of 0.05 V·s^−1^. No redox peaks were observed at the bare GCE electrode (curve a) and NF-GLA/MWCNTs-BSAHFs/GCE electrode (curve c). However, compared with NF-GLA/GOD/GCE (curve b) and NF-GLA/BSA-HFs-GOD/GCE (curve d), NF-GLA/MWCNTs-BSA-HFs-GOD/GCE (curve e) showed a pair of stronger and stable redox peaks on the CVs. It indicated that the redox peaks of NF-GLA/GOD/GCE (curve b), NF-GLA/BSA-HFs-GOD/GCE (curve d), and NF-GLA/MWCNTs-BSA-HFs-GOD/GCE (curve e) were only from GOD. The background current of NF-GLA/MWCNTs-BSA-HFs-GOD/GCE (curve e) was higher than that of NF-GLA/BSA-HFs-GOD/GCE (curve d), which was attributed to the high conductivity of MWCNTs. The anodic and cathodic peak potentials (E_pa_ and E_pc_) of the NF-GLA/MWCNTs-BSA-HFs-GOD-modified GCE were −0.334 V and−0.400 V, respectively, versus Ag/AgCl. The potential difference (∆E) was 0.066 V and the peak current ratio (I_pa_/I_pc_) was close to 1, indicating that the redox reaction of the NF-GLA/MWCNTs-BSA-HFs-GOD/GCE electrochemical process was almost reversible. The formal potential (E°′) of the electrode (E°′ = E_pa_/2 + E_pc_/2) was −0.367 V versus Ag/AgCl. This value is higher than that reported by Li et al. (−0.419 V versus saturated calomel electrode, equivalent to −0.438 V versus Ag/AgCl) [41], and that reported by Cai et al. (−0.438 V versus Ag/AgCl) [42]. The offset of the positive electrode potential is beneficial to promote the efficient biocatalytic [43], which may be due to the weak hydrophobicity of the MWCNT-BSA-HFs composite.

CVs of NF-GLA/MWCNTs-BSA-HFs-GOD/GCE at different scan rates in N_2_-saturated 50 mM, pH 6 phosphate-buffered solution are shown in Figure 5A. In Figure 5B, the relationship between peak current and scan rate was linear in the range of 0.03–0.4 V·s^−1^, and peak current increased by improving the scan rate. This indicates that the reaction is a surface-controlled electrochemical process.

The relationship between the peak potential (E_p_) and the natural logarithm of the scan rate (ln *v*) had two straight lines with slopes of 0.102 and −0.083 (Figure 5C). Using Equation (3) based on Laviron theory in order to obtain k_s_ at a high scan rate (within the range 0.9–1.8 V·s^−1^) [44],
(3)$$E_{p}\;=\;{E^{{}^{\circ}}}^{\prime }+\frac{\mathrm{RT}}{\alpha \mathrm{nF}}\;-\;\frac{\mathrm{RT}}{\alpha \mathrm{nF}}\;\ln v$$
where R is gas constant; T is the temperature (293 K); α, n, F are electron transfer coefficient, number of electrons, Faraday constants, respectively. The value of n, α were calculated 1 and 0.31, and the apparent heterogeneous electron transfer rate constant (k_s_) was calculated as 4.27 s^−1^ using Equation (4), which was greater than previously reported for that of GOD immobilized on graphene (1.96 s^−1^) [45], HFs (2.72 s^−1^) [27], and β-cyclodextrin-MWCNTs-modified electrodes (3.24 s^−1^) [46]. Therefore, the electron migration rate of our biosensor is faster.
(4)$${\mathrm{lnk}}_{s}=\alpha \;\ln(1-\alpha )+(1-\alpha )\ln\alpha \;-\;\ln(\frac{\mathrm{RT}}{\mathrm{nF}v})\;-\;\alpha \frac{(1-\alpha )\mathrm{nF}∆E_{p}}{\mathrm{RT}}$$

The average surface concentration Γ of the FAD of GOD on the GCE surface was estimated to be 2.02 × 10^−9^ mol·cm^−2^ using Equation (5),
(5)$$I_{p}=\frac{n^{2}F^{2}A\Gamma v}{4\mathrm{RT}}$$
where A, R, n, T, and F are the electrode surface area, gas constant, number of electrons, and temperature, respectively. This value was much higher than those of 2.97 × 10^−11^ mol·cm^−2^ at GOD/cobalt sulfide-MWCNTs/NF/GCE [41] and the GOD theoretical Γ value of 1.7 × 10^−10^ mol·cm^−2^ [27], which helped to load more GOD. All the significant improvements can be attributed to the large surface area and enhancement of the electrical signal of MWCNT-BSA-HFs.

Figure 6A showed the CVs of NF/GLA-MWCNTs-BSA-HFs-GOD/GCE in N_2_-saturated 50 mM phosphate-buffered at different pH values. As shown in the Figure 6C, the E_p_°′ was linearly related to pH with the equation of E_p_°′ = −0.0587 pH − 0.0345 (R^2^ = 0.996). The slope value was −58.7 mV·pH^−1^. This value was near to the ideal Nernst value at 25 °C [34], indicating that the electron transfer of GOD immobilized on NF-GLA/MWCNT_S_-HF_S_-BSA-modified GCE electrodes was the process of proton and electron equivalence.

### 3.3. Optimization

Some parameters were studied using CV to optimize the structure of the biosensor and working conditions. Firstly, the effect of the concentration of the GOD solution for NF-GLA/MWCNTs-BSA-HFs-GOD/GCE film formation in the range of 2–12 mg·mL^−1^ was first investigated. Figure 7 shows the cathodic peak current response of the modified electrode with different concentrations of GOD solution using CV. It was observed that modified electrode obtained the best cathodic peak current response when the concentration of GOD solution was 10 mg·mL^−1^. Therefore, this enzyme concentration was used for subsequent biosensor construction.

Subsequently, we investigated the role of supporting electrolytes at different pH values through CV. Figure 6A showed CVs of NF-GLA/MWCNTs-BSA-HFs-GOD/GCE at different pH values. Figure 6B showed that the cathodic peak current increased by raising the pH within the range of pH 3–7, while the current decreased with pH above 7. Since the cathodic peak current intensities of the modified electrodes were similar in phosphate-buffered solution at pH 6 and 7, linear sweep voltammetry (LSV) was used to observe the electrocatalytic behaviors of the modified electrodes in air-saturated phosphate-buffered solution at pH 6 and 7, respectively. Table 1 lists the results. Finally, we used 50 mM phosphate-buffered solution (pH 6) as the condition for subsequent experiments. The pH selected was consistent with the optimal pH 5–6 of GOD (derived from *Aspergillus niger*) [47]. There are two causes: 1. The E°′ of the modified electrode in 50 mM phosphate-buffered solution (pH 6) is greater than that of the modified electrode in 50 mM phosphate-buffered solution (pH 7), which is conducive to promoting efficient biocatalytic reduction [43], and using a lower working potential when detecting Glu in the blood can reduce the interference of the electroactive substances in the blood to the electrodes [30]. 2. Compared with the modified electrode in 50 mM phosphate-buffered solution (pH 7), $K_{m}^{\mathrm{app}}$ of the modified electrode in 50 mM phosphate-buffered (pH 6) is smaller.

Later, the electrochemical methods for detecting Glu were investigated. LSV and differential pulse voltammetry (DPV) were performed on the NF-GLA/MWCNTs-BSA-HFs-GOD/GCE for determination of Glu in the presence of air-saturated phosphate buffer. We found that using DPV could obtain higher current sensitivity and lower $K_{m}^{\mathrm{app}}$ (Table 2), so we chose DPV to detect Glu.

### 3.4. Electrocatalytic Behaviors

The DPV of NF-GLA/MWCNTs-BSA-HFs-GOD/GCE in the presence of Glu in air-saturated 50 mM phosphate-buffered solution was studied. The electron transfer process between the electrodes of GOD in air-saturated phosphate-buffered solution is shown Equation (6). The enzyme-catalyzed reaction of GOD with Glu when Glu was added can be shown by Equation (7). Figure 8A shows that the cathodic peak value of NF-GLA/MWCNTs-BSA-HFs-GOD/GCE decreased with the increase of glucose concentration. This is because after adding Glu into the air-saturated phosphate-buffered solution, the GOD (FAD) on the electrode surface is reduced [48].
(6)$${\mathrm{GOD}\;(\mathrm{FAD})+e}^{-}{+H}^{+}\;↔\;\mathrm{GOD}\;(\mathrm{FADH})$$
(7)Glu+GOD (FAD) → Gluconolactone+GOD (FADH)

Calibration of Glu concentration ranges from 0.01 to 3.5 mM, and the statistical analysis of the cathodic peak current difference (∆I) versus the concentration showed three linear ranges (Figure 8B). ∆I increased linearly by raising Glu concentration from 0.01 to 0.05 mM, with the linear regression equation was ∆I_a_ (μA) = 11.8 C _Glu_ (mM), and ∆I increased linearly by increasing Glu concentration from 0.1–0.9 mM, and the equation was ∆I_b_ (μA) = 2.3185 C _Glu_ (mM) + 0.5788. As well as ∆I increasing linearly with the increase of Glu concentration from 1–3.5 mM, the linear regression equation was ∆I_c_ (μA) = 0.5981 C _Glu_ (mM) + 2.1380. The low detection limit (LOD) was 17 μM (3 S_0_/S, S_0_ and S are respectively the standard deviation measured under blank condition and the slope of calibration curve), which is lower than that reported by Nashruddin et al. (65 μM) [49], Ge et al. (42 μM) [50], and Lin et al. (70 μM) [51]. The sensitivity of the biosensor was 167 μA·mM^−1^·cm^−2^, as shown in Table 3; this value was much higher than 56.12 and 8.5 μA·mM^−1^·cm^−2^ reported in Barathi et al. [52] and Chen et al. [48]. According to the electrochemical version of Lineweaver-Burk, the $K_{m}^{\mathrm{app}}$ can be calculated to be 0.119 mM (Figure 8C), lower than the results of most other biosensors of GOD [27,42,53]. The lower $K_{m}^{\mathrm{app}}$ value indicates that the modified electrode has a strong binding ability with the substrate, showing that this biosensor has a strong affinity for Glu [43]. The improved performance and affinity of the biosensor may be due to the synergistic effect of MWCNT-HFs-BSA and the improved microenvironment of GOD.

### 3.5. Anti-Interference Ability, and Stability of Biosensor

Several electroactive substances commonly found in blood such as vitamin B_1_ (Vit B_1_), vitamin C (Vit C), and sodium chloride (NaCl) were introduced in phosphate-buffered solution to evaluate the anti-interference performance of this biosensor. Considering that the concentration of Glu in human blood is at least 30 times higher than that of physiological interfering substances [54,55], the effects of interfering substances were evaluated by adding 1 mM Glu solution, 0.1 mM Vit B_1_, Vit C, and NaCl solutions and 0.5 mM Glu solution to air-saturated phosphate-buffered solution and observing the signal intensity of the modified electrode by LSV method. The cathodic peak current hardly change significantly when the interfering substance was added, while the addition of two Glu solutions with different concentrations caused a strong cathodic peak current signal (Figure 9 inside the circle). The cathode peak current could not accurately describe the anti-interference ability of the modified electrode because the concentration of Glu solution and interfering substance is different. In order to intuitively study the anti-interference ability of the biosensor, we introduced an interference signal (IS) (Equation (8)) [30]. The IS values of Vit C, Vit B_1_, and NaCl were calculated to be 0.93%, 1.45%, and 3.95%, respectively. The IS value of the biosensor with strong anti-interference ability is low. It can be seen that the biosensor has good selectivity, which is ascribed to the low working potential of this biosensor and the outer layer protection containing NF.
(8)$$\mathrm{IS}\;(\%)=\frac{i_{G+I}-i_{G}}{i_{G}}\;\times \;100\%$$

Here, i_G_ and i_G+I_ represent the response current to Glu in the absence and presence of interference, respectively.

The stability of the modified electrode was researched by CV method (Figure 10A). After 100 cycles, the cathodic peak of the modified electrode only decreased 3%. Additionally, the storage stability of the biosensors was evaluated using the CV method (Figure 10B). We found that the percentage of cathodic peak current only decreased 3% when the biosensor was preserved at 4 °C for 120 days. It shows that this biosensor has excellent storage stability.

### 3.6. Determination of Glu in Plasma

The practical ability of the proposed biosensor to detect Glu in plasma samples was evaluated. The plasma sample was prepared as described above. The rat blood Glu concentration was predefined using a commercial glucose meter (ACCU-CHEK Instant, Roche Diabetes Care GmbH, Jiangsu, China) as 8.2 mM. At present, plasma is usually used to measure Glu concentration in clinical practice, and the level of Glu in plasma is usually 10–15% higher than that in whole blood [56]. Therefore, the plasma Glu concentration of rats after correction was 9.3 mM using Equation (9). Since some interfering substances in real blood samples may cover the electrode surface and obstruct electron transfer, the accuracy of the test results may be reduced. Plasma was diluted with phosphate-buffered solution 310, 62, 18.6, and 12.4 times to obtain plasma with different Glu concentrations, and the recovery rate with biosensors was measured to reduce such adverse effects and experimental errors and improve the accuracy of blood Glu concentration measurement. From Table 4, the recoveries of this method were 95.9–103.9 %. It can be seen that the proposed sensor shows a satisfactory recovery of Glu in plasma samples, validating that the biosensor has potential good practicability in the detection of real samples.
(9)$${\left[\mathrm{Glu}\right]}_{\mathrm{blood}}{=[\mathrm{Glu}]}_{\mathrm{plasma}}\;/\;1.14$$

## 4. Conclusions

In this work, GOD was immobilized on MWCNTs-BSA-HFs composites, and an NF-GLA composite membrane was used to prevent the leakage of immobilized GOD, and a new and simple Glu biosensor was constructed. The composite MWCNTs-BSA-HFs has high specific surface area and good conductivity and biocompatibility, thus promoting the efficient biocatalytic and improving the sensitivity and the performance of the biosensors so that the proposed biosensors possess a wide linear range, low detection limit, high sensitivity and catalytic activity, and good selectivity and stability. In addition, this biosensor has successfully determined Glu concentration in plasma samples with satisfactory recovery. Therefore, the biosensor has great application potential in clinical blood glucose detection.

## Figures and Tables

**Figure 1 sensors-23-03209-f001:**
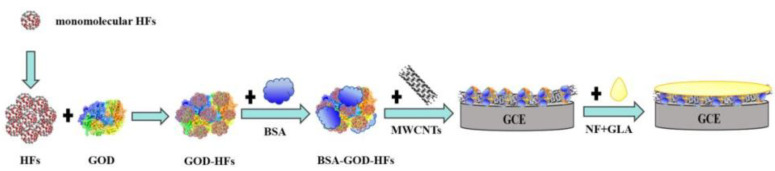
Preparation process of modified electrode.

**Figure 2 sensors-23-03209-f002:**
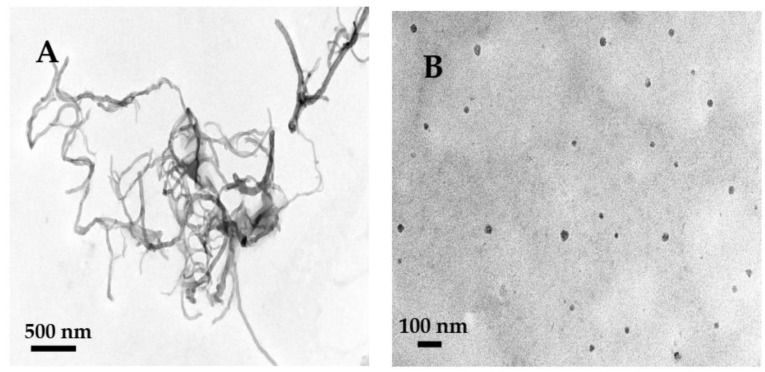
TEM images of MWCNTs (**A**), HFs (**B**).

**Figure 3 sensors-23-03209-f003:**
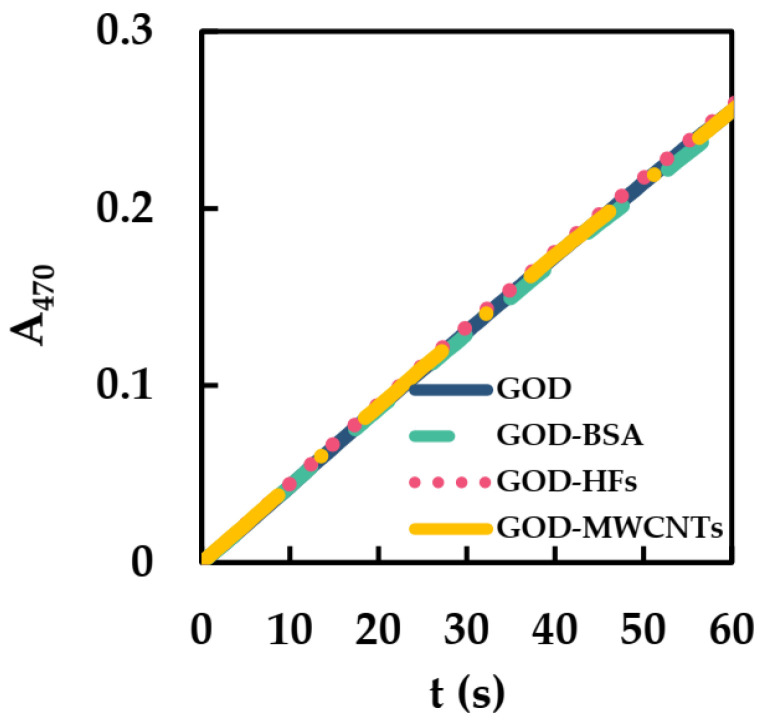
ICRR of GOD (1.20 × 10^−5^ mM), MWCNTs/GOD (1.34 × 10^−3^ mg·mL^−1^ and 1.20 × 10^−5^ mM), HFs/GOD (1.34 × 10^−3^ mg·mL^−1^ and 1.20 × 10^−5^ mM), and BSA/GOD (8.00 × 10^−6^ mM and 1.20 × 10^−5^ mM) in 50 mM phosphate-buffered solution (pH 6, 25 °C) containing HRP (2.50 × 10^−5^ mM), guaiacol (3 mM), and Glu (50 mM).

**Figure 4 sensors-23-03209-f004:**
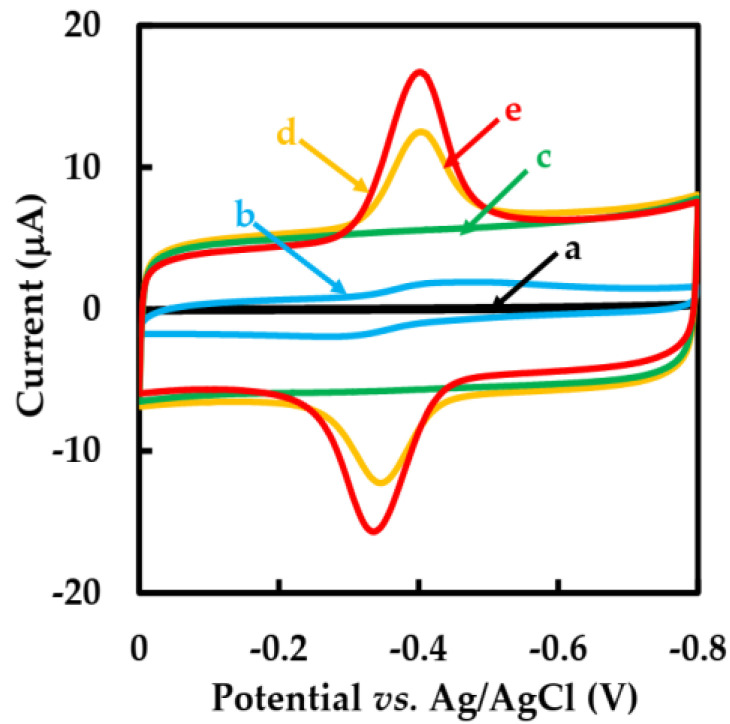
CVs of different modified GCEs: (**a**) bare GCE, (**b**) NF-GLA/GOD/GCE, (**c**) NF-GLA/MWCNTs-BSA-HFs/GCE, (**d**) NF-GLA/BSA-HFs-GOD/GCE, and (**e**) NF-GLA/MWCNTs-BSA-HFs-GOD/GCE in N_2_-saturated 50 mM phosphate-buffered solution (pH 6) at a scan rate of 0.05 V·s^−1^.

**Figure 5 sensors-23-03209-f005:**
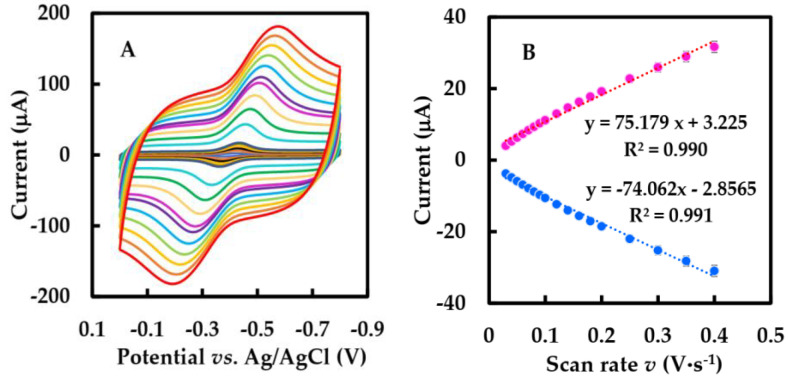
(**A**) CVs of the NF-GLA/MWCNTs-BSA-HFs-GOD/GCE in N_2_-saturated 50 mM phosphate-buffered solution (pH 6) at different scan rates (from inner to outer): 0.01, 0.02, 0.03, … 1.8, 2 V·s^−1^, respectively. (**B**) The relationship of the scan rates (0.03–0.4 V·s^−1^) versus the peak currents. (**C**) The relationship of the peak potential (Ep) versus ln *v*.

**Figure 6 sensors-23-03209-f006:**
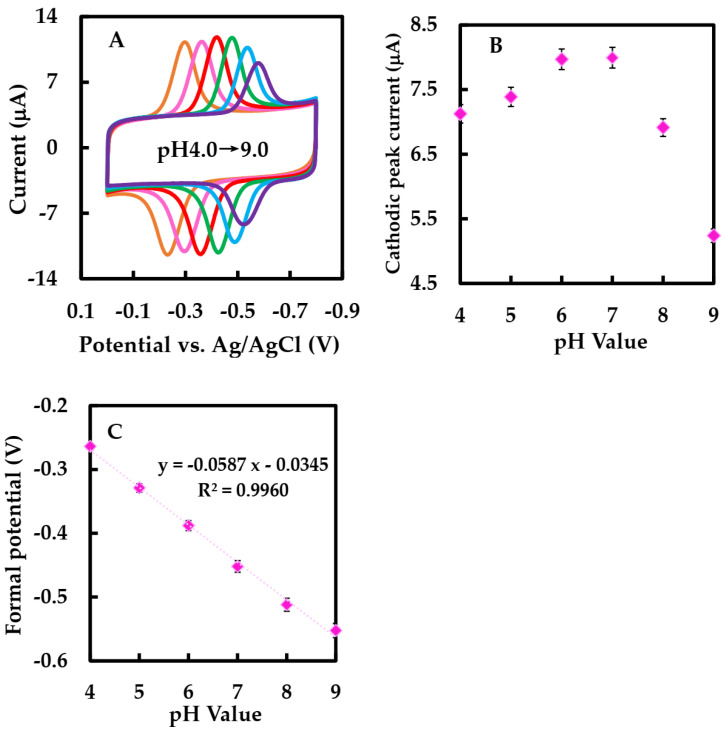
(**A**) CVs of NF-GLA/MWCNTs-BSA-HFs-GOD/GCE electrode in N_2_-saturated 50 mM phosphate-buffered solution at different pH values (from left to right): 4.0, 5.0, 6.0, 7.0, 8.0, and 9.0, respectively, at a scan rate of 0.05 V·s^−1^. (**B**) Plot of cathodic peak current versus pH value. (**C**) Plot of E°′ versus pH value.

**Figure 7 sensors-23-03209-f007:**
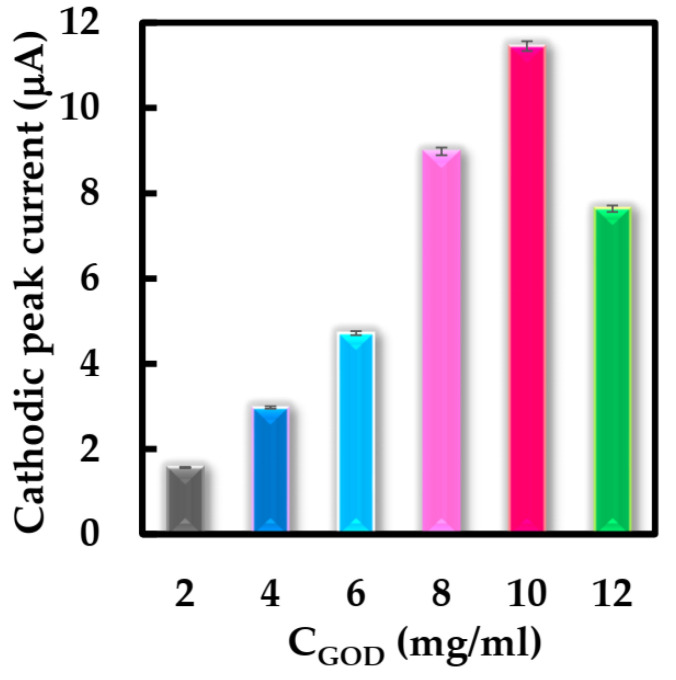
Peak cathodic current obtained from NF-GLA/MWCNTs-BSA-HFs-GOD/GCE with different concentrations of GOD in N_2_-saturated 50 mM phosphate-buffered solution (pH 6) at a scan rate of 0.05 V·s^−1^.

**Figure 8 sensors-23-03209-f008:**
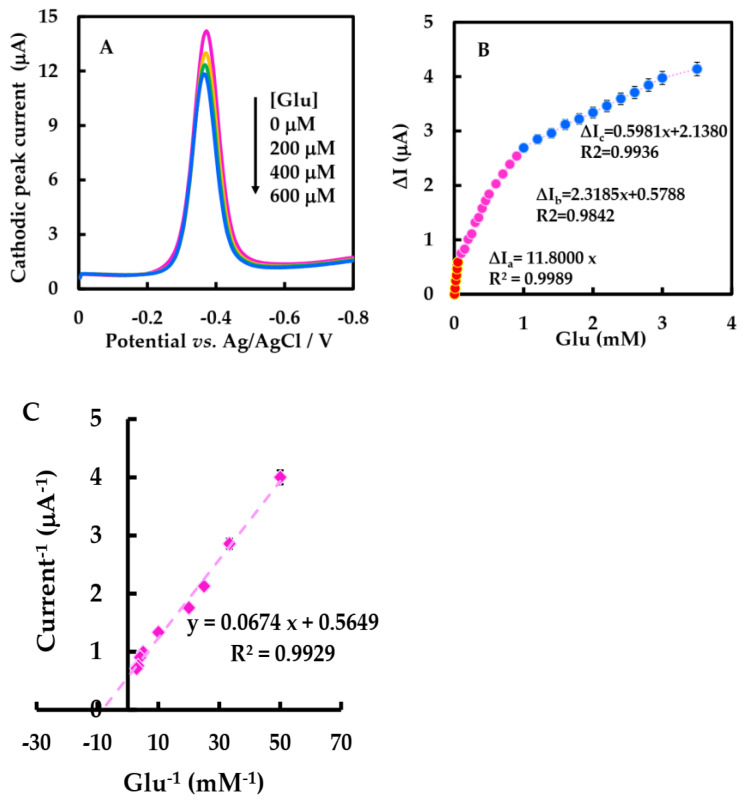
(**A**) DPVs of the NF-GLA/MWCNTs-BSA-HFs-GOD/GCE in air-saturated 50 mM, pH 6 phosphate-buffered solution at a scan rate of 0.05 V·s^−1^ in the absence and presence of different concentrations of Glu (0, 0.2, 0.4, 0.6 mM). (**B**) Linear range from 0.01 mM to 3.5 mM. (**C**) Lineweaver-Burk plot for $K_{m}^{\mathrm{app}}$ determination.

**Figure 9 sensors-23-03209-f009:**
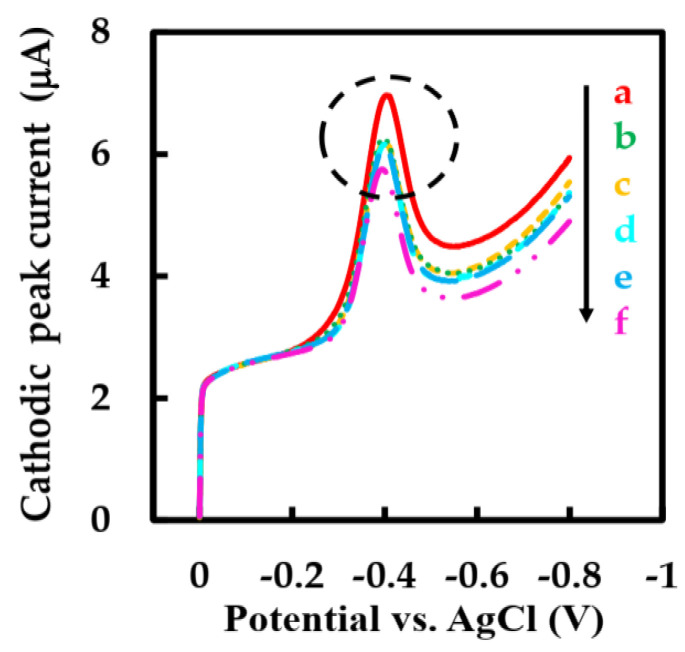
LSVs of NF-GLA/MWCNTs-BSA-HFs-GOD/GCE in blank phosphate-buffered solution (**a**), 1 mM Glu solution (**b**), and Glu solution containing 0.1 mM Vit B_1_ (**c**), Vit B_1_ + Vit C (**d**), Vit B_1_ + Vit C + NaCl (**e**), and 1.5 mM Glu solution (**f**) in air-saturated 50 mM phosphate-buffered solution (pH 6) at a scan rate of 0.05 V·s^−1^.

**Figure 10 sensors-23-03209-f010:**
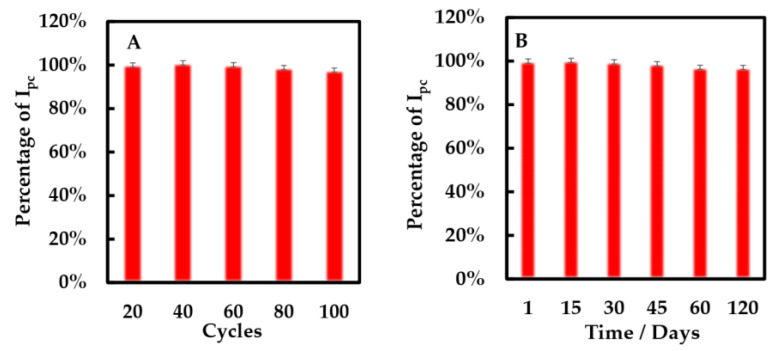
Percentage of cathodic peak current of NF-GLA/MWCNTs-BSA-HFs-GOD/GCE versus (**A**) cycles and (**B**) storage time (days) in 50 mM phosphate-buffered solution (pH 6) at a scan rate of 0.05 V·s^−1^.

**Table 1 sensors-23-03209-t001:** Comparison of Glu-sensing performance of NF-GLA/MWCNTs-BSA-HFs-GOD/GCE in 50 mM phosphate-buffered solution at different pH levels.

pH	E°′ (V)	Linear Range (mM)	LOD (μM)	$K_{m}^{\mathrm{app}}$ (mM)
6	−0.367	0.01–1.6	18	0.645
7	−0.452	0.05–1.1	57	0.955

**Table 2 sensors-23-03209-t002:** Comparison of Glu-sensing performance of NF-GLA/MWCNTs-BSA-HFs-GOD/GCE in 50 mM phosphate-buffered solution using different methods.

Methods	Linear Range (mM)	LOD (μM)	$K_{m}^{\mathrm{app}}$ (mM)	Sensitivity (μA·cm^−2^·mM^−1^)
LSV	0.01–1.6	18	0.645	18
DPV	0.01–3.5	17	0.119	167

**Table 3 sensors-23-03209-t003:** Electrochemical parameters of the electrodes recently modified for Glu detection.

Working Electrode	Liner Range (mM)	LOD (μM)	$K_{m}^{\mathrm{app}}$ (mM)	Sensitivity (μA·mM^−1^·cm^−2^)	Reference
MPC-CHI-GOD/SPCE	0.25–3	4.1	2.1	56.12	[52]
NF/GOD/IL/mPEG-fMWCNTs/GCE	0.02–0.95	0.2	0.143	-	[34]
CHI/GOD-HFs/GCE	0.05–0.5	5	0.694	-	[27]
GOD/PEDOT/CF	0.5–15		6.5	8.5	[48]
GOD/CoS-MWCNTs/NF/GCE	0.008–1.5	5	-	14.96	[41]
PEDOT:PSS/Ti_3_C_2_/GQD-GOD/SPCE	0–0.5	65	-	21.64	[49]
GOD/Au@C/TiO_2_/FTO	0.1–1.6	42	-	29.76	[50]
FTO-CNTS/PEI-GOD	0.07–0.7	70	-	63.38	[51]
GOD/PEDOT:PSS/CNTF	0.05–0.5	43.52	1.63	43.52	[42]
GOD/PtNPs@NG/NF/GCE	0.005–1.1	2	0.66	20.31	[53]
NF-GLA/MWCNTs-BSA-HFs-GOD/GCE	0.01–3.5	17	0.119	167	This work

MPC: mesoporous carbon; CHI: chitosan; SPCE: screen-printed carbon electrode; IL: ionic liquid; mPEG: aminated polyethylene glycol; fMWCNTs: carboxylic acid functionalized multi-walled carbon nanotubes; PEDOT: poly (3,4-ethylenedioxythiophene); CF: carbon fiber; Cos: cobalt sulfide; Ti_3_C_2_: titanium carbide; GQD: graphene quantum dots; FTO: F-doped SnO_2_; CNTs: carbon nanotubes; PEI: polyethylenimine; PEDPT:PSS: poly(3,4-ethylene dioxythiophene): poly (styrene sulfonate); CNTF: carbon nanotube fiber; PtNPs: *platinum* nanoparticles; NG: nitrogen-doped graphene.

**Table 4 sensors-23-03209-t004:** Determination of Glu in plasma samples.

Sample Number	Glu Found by Commercial Glucose Meter (μM)	Glu Found by Modified Biosensor (μM)	Recovery (%)	RSD
1	30	29.6	98.7	1.25
2	150	155.7	103.9	2.30
3	500	479.2	95.9	2.80
4	750	750.3	100.0	2.96

## Data Availability

Not applicable.

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
