# Peer review of "Glucose Biosensor Based on Glucose Oxidase Immobilized on BSA Cross-Linked Nanocomposite Modified Glassy Carbon Electrode"

_sensors, 2023, doi:10.3390/s23063209_

Round 1
Reviewer 1 Report
The manuscript “Glucose biosensor based on glucose oxidase immobilized on BSA cross-linked nanocomposite modified glassy carbon electrode” describes the fabrication of an enzymatic glucose biosensor using different components. This topic has been frequently studied in the literature, hence the authors need to highlight the novelty of the present work. Although it was mentioned that biosensor is simple, using more than 5 components for fabricating the modified electrode seems to be a complicated process.
The following comments can be regarded for improving the manuscript.
1- In the abstract, it was mentioned that TEM was used for analyzing the modified materials, however, the TEM result is related to the pure MWCNTs. Moreover, in line 127, it was stated the MWCNTs is uniform, hollow and curved. Whether these interpretations can be deduced from the TEM graph with a scale of 500 nm?
2- In table 1, ICRR is the same for all components. Do all the components have the same effect?
3- In line 172, does the value – 0.367 larger than – 0.419?
4- How did the authors select the papers in table 4? There are a lot of papers in the literature that has better parameters than the present work.
5- For evaluating interferences, the chronoamperometry technique (current vs. time) is usually used. Why do the authors utilize the CV technique? Moreover, the result in table 5 should adequately be explained.
6- There are typos in the text.
7- The figure captions should be corrected.
Reviewer 2 Report
Great study. Innovative way of using materials to improve sensitivity and performance of biosensors.
This research will push the glucose detection topic one step closer for wide-scale use in clinical application.
Author Response
Many thanks for your kindly review and support.
Reviewer 3 Report
Article “Glucose biosensor based on glucose oxidase immobilized on BSA cross-linked nanocomposite modified glassy carbon electrode” is devoted to the development of a sensor for determining the level of glucose in the blood. The study is topical and corresponds to the theme of the journal.
I would recommend that the authors pay attention to the following comments:
1. I think that the authors should to formulate the purpose of the study in the introduction clearly, and not just list the work done.
2. Tables 1 and 5 are not informative. They can be replaced by one corresponding sentence in the text.
3. Equations (3)-(5) require a detailed interpretation of the quantities included in them.
4. Figures 5 and 6 are too bulky. I offer fig. 5b and fig. 6 b makes separate figures.
5. The data in table 6 require the specification of uncertainties. Without these data, it is difficult to talk about the performance of the developed sensor.
I believe that after corrections and improvements, the article can be published.

Reviewer 4 Report
The manuscript " Glucose biosensor based on glucose oxidase immobilized on BSA cross-linked nanocomposite modified glassy carbon electrode" was submitted to Sensors, which describes composite material to establish glucose biosensor.
From my point of view, there are no serious concerns or obstacles in this review that need to be further clarified and revised by the authors. According to my opinion, this manuscript is logically organized, well referenced, and represents a valuable contribution. However, minor changes have to take into account:.
Q1. The novelty of the work should be emphasized in the introduction section.
Q2. The error range and relative standard deviation should be provided in the experiment of real plasma samples.
Q3. The manuscript also should undergo a round of proofreading to correct language and proofreading errors.
Round 2
Reviewer 1 Report
The revised manuscript “Glucose biosensor based on glucose oxidase immobilized on BSA cross-linked nanocomposite modified glassy carbon electrode” was evaluated.
Although the manuscript was improved, the below comments should be discussed:
· In response to comment 2, the authors stated that BSA, MWCNTs, and HFs had the same effect on the catalytic activity of glucose oxidase and could not affect the catalytic activity of glucose oxidase. However, they didn’t respond to "why the different species have the same effects". Instead, they delete the ICRR data and related text.
· In comment 3, the absolute value of -0.367 is smaller than - 0.419 and it is important to regard the absolute value.
· About comment 4, I didn’t convince how did the authors select the papers for comparison? I think the papers should have something in common. (e.g. they can have a common component in their working electrodes)
· In comment 5, I think for evaluating interferences the chronoamperometry technique should be used since the peak changes in the CV diagram is not clear and explicit. Moreover, do you have any references for equation 8?
Reviewer 3 Report
The authors took into account most of these comments. I think that the article can be published.
Round 3
Reviewer 1 Report
The manuscript has improved and it can be accepted for publication in sensors.